# Predictors of Adaptive Behaviors in Individuals on the Autism Spectrum as Assessed by Teachers and Parents: An Analysis Based on ABAS-3

**DOI:** 10.3390/jcm13247607

**Published:** 2024-12-13

**Authors:** Janusz Kirenko, Anna Prokopiak, Maciej Wodziński

**Affiliations:** 1Department of Pedeutology and Health Education, Institute of Pedagogy, Faculty of Pedagogy and Psychology, Maria Curie-Skłodowska University, Głęboka 43, 20-612 Lublin, Poland; janusz.kirenko@mail.umcs.pl; 2Department of Special Psychopedagogy, Institute of Pedagogy, Faculty of Pedagogy and Psychology, Maria Curie-Skłodowska University, Głęboka 43, 20-612 Lublin, Poland; anna.prokopiak@mail.umcs.pl; 3Department of Ontology and Epistemology, Institute of Philosophy, Faculty of Philosophy and Sociology, Maria Curie-Skłodowska University, Maria Curie-Skłodowska Sq. 4, 20-031 Lublin, Poland

**Keywords:** adaptive behaviors, ABAS-3 (Adaptive Behavior Assessment System), autism spectrum disorder (ASD), diagnosis, diversity of perspectives (parents, teachers)

## Abstract

**Objectives:** This present study focuses on analyzing the adaptive behaviors of individuals on the autism spectrum as perceived by parents and teachers of these individuals. **Methods:** This study was conducted in Poland with the use of the ABAS-3 (Adaptive Behavior Assessment System). The ABAS-3 tool involves both parents (or primary caregivers) and teachers in the diagnostic process and monitoring of adaptive development. The study included 99 individuals (29 girls, 70 boys) aged between 5 and 21 years. **Results:** The analysis of the results showed statistically significant discrepancies in the perception of adaptive skills diagnosed as assessed by parents and teachers. Furthermore, differences were found in the predictors of the General Adaptive Composite and adaptive domains. **Conclusions:** The results indicate the complexity of the assessment of adaptive skills by a parent of a child with autism spectrum disorder, as well as a teacher, and the need to include different perspectives in the process of diagnosing and supporting individuals with ASD.

## 1. Introduction

### 1.1. Background of the Study

In recent decades, adaptive behavior has become particularly important in diagnosing and understanding autism spectrum disorder (ASD) [1,2]. The term refers to learned conceptual, social, and practical skills used by people in everyday life, and is a continuation of the historical approach that attaches importance to adaptive behaviors in the diagnosis of developmental delays/intellectual disabilities [3]. Low levels of adaptive skills are also found among individuals with ASD without cognitive delays. It is assumed that more than 60% of adults on the autism spectrum have low levels of these skills [1]. Modern tools for assessing adaptive behaviors are additionally often crucial in diagnosing ASD and tailoring individualized therapeutic support. The most commonly used include the Vineland Adaptive Behavior Scales (VABS) [4], sometimes qualitative research [1], and others such as, e.g., the Inventory for Service Planning and Individual Programming (ICAP) [5]. It should be noted, however, that they lack explicitness and consistency in the assignment of individual predictors of adaptive behaviors due to the type of subject conducting the assessment, i.e., the parent or the teacher. Compared to other tools, the ABAS-3 (Adaptive Behavior Assessment System), proposed here, is of particular value because of the involvement of parents and teachers in the diagnostic process, which—given few research reports regarding theoretical and empirical analyses that verify inquiries into predictors of adaptive behaviors by analyzing the relationship between parents’ ratings and teachers’ ratings, where, as stated by Moore et al. [6], parents and teachers have different perceptions of the adaptive abilities of children with autism—is becoming increasingly important, and for this reason, both groups should be considered when planning personalized interventions and support.

Also, the Vineland Adaptive Behavior Scales tool is only partially translated and pre-adapted in Poland. Due to the advanced adaptation process of the ABAS-3, this tool was chosen. The main limitation of the ABAS-3 tool currently in Poland is its adaptation only to 20-year-old subjects and the lack of studies of subjects on the autism spectrum. Comprehensive measures are only possible when the disability assessment system or the system for developing strategies for working with a student on the autism spectrum are compatible with each other, at least to the extent that the American system provides for it. It is to be hoped that the ABAS-3 issued in Poland will be extended with a full adaptation as soon as possible and that the Polish health, education and social welfare systems will use the same jointly developed tools for diagnosis and planned support. Only then will the ABAS-3 be fully adapted to the purpose for which the assessment is being prepared, and the results obtained will be useful and properly applied. The original tool offers such possibilities but requires appropriate adaptation for people on the autism spectrum.

To date, studies of the adaptive behavior of individuals with ASD have rarely analyzed the relationship between parent and teacher ratings [7]. Where such analysis has been undertaken, research has typically been conducted on relatively small samples [8]. Assessments by different groups of informants are considered the gold standard in the diagnosis and assessment of children’s functioning. Despite this, few studies have focused on examining the concordance of assessments of adaptive functioning in adolescents with ASD, particularly using the ABAS-3 tool. Our study, conducted on a large and diverse group of adolescents with ASD, examined the concordance of parent and teacher assessments of both adaptive functioning and ASD-specific symptoms at different time points [9].

### 1.2. Objectives of the Study

The purpose of this study is to extract the areas of individual adaptive behaviors from among those proposed by the ABAS-3 as predictors of the General Adaptive Composite (GAC) and the three adaptive domains (conceptual, social and practical) of the studied children and adolescents on the autism spectrum as assessed by parents and teachers.

Our study addresses the important question as to whether there are differences in the areas of individual adaptive behaviors as predictors of the General Adaptive Composite (GAC) and the three adaptive domains (conceptual, social and practical) of children and adolescents on the autism spectrum, as assessed by parents and teachers. It provides a comparative analysis of parent and teacher ratings of adaptive behaviors in individuals with autism spectrum disorder (ASD) using the Adaptive Behavior Assessment System, Third Edition (ABAS-3), therefore contributing to the existing literature by demonstrating how the ABAS-3 can be used to analyze adaptive behaviors in different contexts and by illustrating the variability in assessments between key stakeholders. This study highlights discrepancies between parent and teacher ratings across various adaptive domains and skills, which is currently an unfilled gap in this area, emphasizing the need to incorporate multiple perspectives for a holistic understanding of adaptive behaviors in ASD. Due to the fact that it was conducted in Poland, the research underscores cultural and methodological challenges in using the ABAS-3 for assessing individuals with ASD, revealing the limitations of existing tools and the need for localized adaptations.

Due to the diagnostic nature of the research problem, no hypotheses were formulated.

## 2. Literature Review

### 2.1. Autism and Adaptive Behaviors

The modern understanding of the concept of adaptive behavior has become relevant since it was incorporated into the definition of intellectual disability by Luckasson et al. [3], sparking heated debate and controversy over its meaning and application. Adaptive behavior refers to a set of daily activities and skills necessary for social communication and interaction and is now often considered a key element in the diagnosis and understanding of the autism spectrum [10].

Autism spectrum disorders (ASD; ICD-10: pervasive developmental disorder, PDD) are characterized by persistent deficits in social communication. Individuals with ASD have a variety of difficulties in understanding social contexts and adapting their behavior to them. Although the etiology of ASD is not yet completely understood, it is now most commonly described as a combination of genetic predisposition and environmental influences. The autism spectrum ranges widely among individuals—from those with profound intellectual disabilities, not using verbal communication, with low levels of adaptive skills, to those with good or even very good cognitive and language skills, who have good chances for independent living and employment [11,12].

Surprisingly, however, low levels of adaptive skills are also often found in people with ASD who do not exhibit cognitive or language delays. Despite the potential to live on their own, most adults with autism do not achieve satisfactory outcomes in the areas of employment, independence and interpersonal relationships. Even with normal cognitive and language functions, more than 60% of people with autism show low levels of adaptive skills [1]. This underscores the importance of a standardized assessment of adaptive skills in the ASD diagnosis process.

Current Adaptive Behavior Assessment tools play a key role in accurately identifying ASD, as well as in determining appropriate forms of personalized support [5]. They enable a better understanding of the complex picture of autism and tailor support to the individual needs and abilities of each person on the spectrum.

### 2.2. Predictors of Adaptive Behaviors in Autism

Research on predictors of adaptive behaviors in individuals with autism spectrum disorder (ASD) reveals different findings, reflecting the complexity of this domain. Some studies indicate that variables such as age, gender, family income and years of formal education are not significant predictors of adaptive behaviors [7,13]. By contrast, other studies highlight age and gender as significant factors influencing deficits in daily living skills [14].

Age emerges as a particularly strong predictor of communication and social skills [15], while motor skills are consistently identified as key predictors of adaptive behaviors [16,17]. Differences in motor coordination, particularly hand coordination, may underlie variations in adaptive functioning [16]. Furthermore, intelligence quotient (IQ) has been found to be a poor predictor of adaptive functioning in children with ASD, who often score lower in adaptive behaviors than their IQ would suggest [2]. Notably, adaptive skills in children with ASD tend to decline with age, suggesting a developmental trajectory distinct from typically developing peers [13].

Additionally, early diagnosis of executive function deficits has been identified as a potential predictor of later adaptive behaviors, underscoring the importance of early intervention [15]. Mental age in infancy also correlates with trajectories in communication, daily living skills and socialization [18].

### 2.3. The Role of Parents and Teachers

Studies exploring adaptive functioning often highlight the discrepancies between parent and teacher ratings, revealing context-dependent variations. For example, a study using the ABAS-3 found that teachers consistently provided higher adaptive behavior ratings compared to parents, suggesting better functioning in structured school environments [19]. These discrepancies may reflect the differing perspectives and expectations of parents and teachers or variations in behavior across home and school settings. Studies indicate that while parent and teacher ratings are moderately correlated, teachers often rate adaptive skills higher than parents. For instance, Stevens et al. [20] found that teachers rated youth with ASD and intellectual disabilities as having higher adaptive skills compared to parent ratings, with autism symptomatology significantly predicting these discrepancies

While there is limited research directly comparing parent and teacher ratings, existing studies suggest that differences in scores could indicate unique adaptive behavior profiles in various contexts. Understanding these variations is crucial for tailoring interventions that address both home and school environments [21].

### 2.4. The Use of ABAS-3

The ABAS-3 has been increasingly used in assessing adaptive behaviors in individuals with ASD, although it was not specifically designed for this population. Compared to the Vineland Adaptive Behavior Scales (VABS), the ABAS-3 offers a broader focus on daily living and practical skills but may lack specificity in assessing the unique challenges faced by individuals with ASD. For example, while the VABS is frequently used to measure adaptive behavior trajectories, particularly in large-scale studies (e.g., Ref. [18]), the ABAS-3 provides a more standardized framework for examining differences across informants, such as parents and teachers.

However, the limitations of using the ABAS-3 in ASD populations should be addressed. The tool’s general design may introduce biases, as it does not account for the distinct developmental profiles of individuals with ASD. For instance, the developmental decline in adaptive skills with age, as observed in ASD populations [13,15], may not be fully captured by the ABAS-3’s scoring system. Future studies should consider these limitations and explore how the ABAS-3’s findings align with those derived from the VABS to ensure comprehensive assessments.

In the Polish context, it is worth noting that until the 1990s, professionals in Poland had very limited access to materials and tools for diagnosing disabilities, including those related to adaptive behaviors. The only tools used in Poland until recently for assessing adaptive behaviors were B. Doll’s Social Maturity Scale and Witkowski’s PAC (Progress Assessment Chart). For this reason, work on adaptations and the promotion of the use of new tools in this context is so important.

## 3. Methods

### 3.1. Study Participants

As part of the initial work on the adaptation of the ABAS-3 in Poland, a group of individuals on the autism spectrum was included in this study. All subjects had a diagnosis of autism according to the ICD-10 criteria, which is still in force in Poland. According to these criteria, the severity of symptoms is not specified. The age range is clearly defined for the ABAS tool, so chronological age was decisive in the selection of study participants. The study participants came from 28 different centers and educational facilities, including integrated and special kindergartens, special education centers and integration schools.

The study included 99 individuals with autism spectrum disorders from Poland, including 29 girls (29.29%) and 70 boys (70.71%). The age of the studied children and adolescents ranged from 5 to 20 years, with an average age of 10.89 years—12.31 years for girls and 10.32 years for boys.

The research was carried out as part of the functional diagnosis of the study subjects at the above-mentioned centers. Both the carers and the subjects themselves were informed that they could withdraw their consent to take part in the study at any time without stating a reason and without any consequences and that all personal data would be anonymized, meaning that the carer’s data and the child’s data could not be linked to the results of the study.

In the study group, more than 30% were only children, while the rest of the study population had at least one sibling (see Table 1).

### 3.2. The Tool Used for the Study

Schalock et al. (following Ref. [22]) identified 200 tests used to examine adaptive behaviors, while elsewhere, only four were considered valid [23]. Despite attempts to do so, it is difficult to construct a universal instrument that encompasses the needs and demands that can arise from the diversity of activities, environments and life contexts of individuals with ASD [5].

This study uses the ABAS-3 (Adaptive Behavior Assessment System) tool, developed by Patti Harrison and Thomas Oakland. The ABAS-3 is a comprehensive standardized system for assessing adaptive skills, used in a variety of social settings and age groups. It should be emphasized that the ABAS-3 was not specifically designed to assess individuals on the autism spectrum.

The tool consists of five forms that analyze adaptive skills. The forms differ to match different age groups and contexts. Two are addressed for parents or primary caregivers and two for teachers or other caregivers of the child, emphasizing the role of both parents and teachers in the process of diagnosing and monitoring adaptive development.

The role of parents is particularly prominent in Forms 1 and 2, where they assess the adaptive skills of newborns, infants, preschool children and children and adolescents up to 21 years of age at home and in other familiar settings. Teachers have dedicated Forms 3 and 4, where they focus on assessing the adaptive skills of children and adolescents aged 2 to 21 in the context of daycare, preschool and school. The last form is designed for adults.

Scores from the ABAS-3 are used in the diagnosis of developmental disorders, identification of functional limitations, treatment planning and research [24,25]. Classification and assessment are based on three levels: the General Adaptive Composite (GAC), three adaptive domains (conceptual, social and practical) and individual adaptive skill areas [26]. Research also highlights the utility of the ABAS-3 tool in evaluating adaptive functioning in individuals with autism. Tamm, Day and Duncan [27] demonstrate that the ABAS-3 effectively documents adaptive behavior profiles and identifies discrepancies between IQ and real-life skills, making it valuable for diagnosis and therapy planning, while Colantuono et al. [28] underscore its effectiveness in capturing long-term changes in daily functioning across domains like self-care and health in neurodiverse populations, including autism.

The use of the ABAS-3 (Pracownia Testów Psychologicznych Polskiego Towarzystwa Psychologicznego sp. z o. o., Warsaw, Poland) in Poland was preceded by an appropriate adaptation procedure. Standards were developed for the 0–20 age group, taking into account specific contexts and age groups. Currently, work is in progress in Poland to prepare standards for people up to the age of 89.

In summary, the ABAS-3 plays a key role in the assessment and development of adaptive skills, while involving parents and teachers in the assessment process allows for a comprehensive and personalized understanding and support of the individual. The system reflects current standards and is a comprehensive tool used in diagnosis, treatment and research on adaptive behaviors.

### 3.3. Study Procedures

All subjects were diagnosed with holistic developmental disorders, including autism or Asperger’s syndrome (according to ICD-10), and were issued a statement of special education needs (a requirement of the educational system in Poland for additional educational services). The respondents completing the questionnaires (parents and teachers) had the opportunity at each stage of this study to contact the authors of the study; they could ask questions and provide suggestions on the questionnaires.

The interviewers were appropriately trained and had adequate knowledge and experience in basic educational and psychological assessment and test interpretation. There was a person in charge of coordinating the completion of the forms at each facility. Parents, teachers, family members and carers of the subjects and the subjects themselves completed the form using printed questionnaires.

The analysis of the reliability of the ABAS-3 tool in the Polish context was performed by Wojciech Otrębski, Ewa Domagała-Zysk and Agnieszka Sudoł [26] on the basis of the results obtained in the study of the standardization sample. Internal consistency, standard errors of measurement, stability and consistency between different worksheets concerning the same individuals were determined. The reliability coefficient ranged from 0.86 to 0.99 depending on the worksheet, the lowest being in the age range 0;2–0;11. The recalculated scores for adaptive functions correlate moderately strongly with each other.

The following statistical methods were used: the student’s *t*-test for independent pairs and multivariate stepwise analysis.

A *t*-test was performed when the distributions in the populations were assumed to be approximately normal. This is important for small samples (n < 30) because, for larger samples, the test has robust-to-moderate deviations from this assumption. Furthermore, the test requires the assumption of equality of variance in both populations.

Associations between variables were established using stepwise multiple regression analysis, which allows for the arrangement of explanatory variables that are significant in explaining the dependent variable to be identified and the strength of the association between them to be estimated in the regression model so constructed. The essence of stepwise regression is to leave a minimum set of explanatory variables in the regression function model while maximizing the coefficient of determination and minimizing the mean square of the deviations from the regression model in order to identify the dimensions of those variables that most often combine with each other and those that are in conflict with each other, i.e., interacting with the phenomenon under study in a stimulating or inhibiting way. Stimulants are variables with positive regression parameter values, while destimulants are variables with negative parameters. On the other hand, statistically insignificant variables, so-called neutral variables, do not participate in the analysis of the correlation under study. At the same time, multiple regression coefficients have different units of measurement, so they cannot be used to directly compare the strength of the association between individual explanatory variables and the explained variable.

## 4. Results

### 4.1. Summary of Key Findings

Below, we present a summary of the key findings, including a comparison of parent and teacher ratings on key adaptive domains and skills (Table 2 and Figure 1) and key predictors of General Adaptive Composite (GAC) and adaptive domains in the opinion of parents and teachers (Table 3 and Figure 2). A more detailed summary and analysis is presented and discussed below.

### 4.2. Overall Score Analysis

Table 4 and Figure 1 show a comparison of results obtained with the use of the ABAS-3 for children and adolescents with ASD, including parents’ (P) and teachers’ (T) ratings. It presents the main research categories: General Adaptive Composite (GAC), adaptive domains and individual adaptive skill areas (see Table 4).

Comparisons made using the *t*-test for independent pairs revealed significant statistical differences in the levels of assessments of the adaptive skills of the studied children and adolescents performed by the parent and the teacher (see Table 5). A closer analysis of the obtained data indicates that parents, in almost every dimension of the tool used, obtained the highest arithmetic means and the lowest values of standard deviations in the ratings of their children. However, this is most evident—due to the moderate strength of the effect—in the three areas of individual adaptive skills, where parents of adolescents on the autism spectrum rated the skills of their children significantly higher than their teachers: self-direction (to = 4.853; *p* < 0.000), social skills (to = 3.665; *p* < 0.000) and home life (to = 4.994; *p* < 0.000). In two other areas of individual adaptive skills, such as health and safety (to = 4.994; *p* < 0.000) and leisure time (to = 4.994; *p* < 0.000); in two adaptive domains, conceptual (to = 3.366; *p* < 0.001) and practical (to = 3.186; *p* < 0.002); and, most importantly, as in the General Adaptive Composite (GAC) (to = 2.199; *p* < 0.031), the ratings in the parents’ group are also significantly higher than those in the teachers’ group, but the effect is already weak. Furthermore, in one adaptive domain—social (to = 0.486; *p* > 0.628), and in two areas of individual adaptive skills, communication (to = 0.570; *p* > 0.120) and practical learning (to = 0.151; *p* > 0.253), the parents’ average ratings were higher than the teachers’, albeit with a statistically insignificant difference. Only in self-care (to = −0.596; *p* > 0.552) and socializing (to = −1.197; *p* > 0.234) was there a change in their ratings, in favor of teachers, also at a statistically insignificant level.

Table 5 presents a regression analysis showing individual adaptive skills areas as predictors of a General Adaptive Composite (GAC) and the three adaptive behavior domains in the studied population of children and adolescents with ASD as assessed by the parent and the teacher. In the model created in the parents’ group, explaining 85% of the variation in the response variable of General Adaptive Composite (GAC), only two domains of individual adaptive behaviors, including self-care (SC) and leisure time (LT), were found at the level of statistical significance. It is, therefore, reasonable to believe that the increase in the value of this general adaptive disposition of children and adolescents on the autism spectrum is accompanied, according to the assessing parents, primarily by a directly proportional increase in the degree of skills required by them to demonstrate independence, responsibility and self-control, and to engage in and plan leisure time, including recreational activities. By contrast, in the group of assessing teachers, the explanatory variables that were included in the regression model are far more numerous; their level of variation is higher, at 94%, in explaining the response variable of GAC. In addition to the already mentioned areas of self-care and leisure time, the partial indicators at the level of statistical significance include an additional four: self-determination (SD), school life (SL), health and safety (HS) and communication (C). All of these indicators express directly proportional relationships; hence, the intensification of this disposition in this group is accompanied, in addition, by an increase in the skills needed to take care of oneself, attend school and class, take care of one’s health and respond to illness or injury, as well as have the speech and listening skills required to communicate with others. A broader palette of individual adaptive behavior areas among assessing teachers is apparent here, which seems obvious given the tasks facing them in the context of the professionalization of the teaching profession. The situation is different with regard to the assessments made by parents, who strongly subjectivize the image of their child, limiting the areas of individual adaptive behaviors to the broadly understood child’s “well-being”.

However, this specific regularity does not apply to the response variable of the conceptual adaptive domain, responsible for the behaviors needed to communicate with others, use knowledge and manage and bring tasks to completion, where, in both parents’ and teachers’ assessments, among the explanatory variables, at the level of statistical significance, in the two regression models proposed by the parents and teachers surveyed, each included the same four areas of individual adaptive behaviors, namely practical learning (PL), self-determination (SD), communication (C) and socializing (S). They explain ca. 97–98% of the variation in the response variable, and their relationship is directly proportional. Therefore, the analyzed disposition is determined by an increase in the skills needed to function and behave appropriately in society, to interact socially and maintain contact with others and to take care of oneself by, inter alia, developing the speech and listening skills needed to communicate with other people. At the same time, the gradations of separate areas of individual adaptive behaviors in the compared groups are overtly different, representing hierarchical opposites. Within the conceptual domain, the parent strongly emphasizes their child’s practical learning skills and communication, while the teacher—in the context of the same child/student—emphasizes socializing and self-determination.

On the other hand, with regard to the response variable of the social adaptive domain, accumulating the behaviors needed to engage in interpersonal relationships, activities related to social responsibility and the use of leisure time, the created interdependence models in the groups of the assessing parent of the studied child and, at the same time, the assessing teacher of the studied student, are slightly different. Among the explanatory variables at a statistically significant level were, in each group, three indicators of individual adaptive behaviors. In turn, the created models explain between 91%, in the group of the assessing parent, and 76%, in the group of the assessing teacher, of the variation in the response variable. In equal numbers of cases, the nature of these relationships is directly and inversely proportional, where direct proportionality applies in both groups to two areas of individual adaptive behaviors, leisure time (LT) and socializing (S), while inverse proportionality applies to only one area in each group, social skills (SS) in the case of assessing parents and communication (C) in the teachers’ group. Therefore, the implementation of this disposition by the studied children and adolescents on the autism spectrum as assessed by parents and teachers is to the greatest extent determined, and thus reinforced, by the ability to engage in and plan leisure time, including recreational activities, as well as to establish social interaction and maintain contact with others. On the other hand, in contrast to this disposition—in the parent’s view—there are the skills needed to function and behave appropriately in society, such as getting around, showing interest in activities outside the home and recognizing various infrastructure, while in the teacher’s view, these include the area of speech and listening skills needed to communicate with others. The difference between these buffers appears to be significant, thus confirming the interpretive direction adopted here, as discussed earlier.

In turn, among the explanatory variables comprising the interdependence model, four areas of individual adaptive behaviors at a statistically significant level were found in the assessing parents’ group, while in the assessing teachers’ group, there were five such significant areas. Four of them were almost identical, including social skills (SS), self-care (SC), and health and safety (HS); this “almost” refers to the fourth area of behavior, differentiated by the competence of the assessing subject, namely for the parent—home life (HL), and for the teacher—school life (SL). In contrast, the fifth area in the group of assessing teachers concerned leisure time (LT). The created models explain 97% (in the first case) and 98% (in the second case) of the variation in the response variable of the practical adaptive domain, which includes behaviors essential in taking care of personal and health needs at home, in the classroom or in a workplace, as well as to function in society. In all cases, the nature of these relationships is directly proportional. Analogous to the earlier regression models, the implementation of this disposition by the studied children and adolescents with ASD is to the greatest extent determined by their high level of presentation of the skills needed to function and behave appropriately in society; to show independence, responsibility and self-control; and to take care of health and respond to illness or injury, in addition to taking care of one’s home and place of residence or school and classroom, manifested in activities such as cleaning, helping adults with household chores or school work and taking care of one’s own belongings. This set of areas of individual adaptive behaviors is the most coherent among the domains and assessed subjects studied here apart from the “teacher”-specific area of skills needed to engage in and plan leisure time, of course, including recreational activities.

### 4.3. Detailed Conclusions

Based on the data and analysis presented, the following conclusions can be drawn.

Differences in assessments given by parents and teachers: There is a clear difference in parents’ and teachers’ perceptions of the adaptive skills of children and adolescents on the autism spectrum. Parents consistently rate their children on the autism spectrum higher in almost every dimension of adaptive skills, indicating a potential subjective perception of their children’s abilities.

Significant areas of divergence: Statistically significant differences in ratings between parents and teachers are found in the areas of adaptive skills, such as self-determination, social skills, home life, health and safety and leisure time, as well as conceptual and practical adaptive domains.

Converging scores in the conceptual adaptive domain: It is interesting to note that both parents and teachers agreed on the key behavioral areas that have an impact on the conceptual adaptive domain, pointing to communication, practical learning, self-determination and socializing as the main predictors.

The pattern of predictors in the regression models: In the regression model for assessing parents, only two areas—self-care and leisure time—were significant predictors of the General Adaptive Composite (GAC). By contrast, teachers identified a broader range of areas that influence GAC.

Difference in perception of the social adaptive domain: Although both groups identified leisure time and socializing as key predictors of the social adaptive domain, there were differences in perceptions of other areas, such as communication and social skills.

Coefficient of Determination (R2): High R2 values suggest that the model fits the data well, which can increase confidence in forecasts and recommendations based on the model.

Potential implications for educational and intervention practice: Differences in assessments between parents and teachers may have important implications for educational and intervention practice. They may point to the need for more personalized teaching and support strategies that account for the individual strengths and weaknesses of each student on the autism spectrum. In addition, they underscore the importance of communication between teachers and parents to gain a more complete picture of the student’s needs and abilities.

Limitations of the study: Subjectivity in parents’ assessments may result from a natural tendency to view their own children in a more positive light. It is important to account for these differences in interpreting the results and adjust interventions accordingly. The ABAS-3 tool does not consider the adaptive needs of individuals with disabilities, including those on the autism spectrum.

In summary, the analysis indicates significant differences in parents’ and teachers’ perceptions of the adaptive skills of children and adolescents on the autism spectrum. These differences may be crucial for planning and providing educational and intervention support to this population.

## 5. Discussion

### 5.1. Interpretation of Study Results

The results and conclusions presented here provide important insights into parents’ and teachers’ assessment of the adaptive skills of children and adolescents on the autism spectrum.

The ABAS-3 is distinguished by the simplicity of conducting surveys, which is due to the short time required to complete the questionnaire. This is particularly useful when working with large groups (cf. Refs. [29,30]).

The tool also allows for the analysis of differences in assessments stemming from varying data sources, such as a teacher and a parent, or an assessment made by the researcher itself. Such comparisons support clinical practice and therapeutic intervention by providing insights into the diverse perspectives of the examinee’s environment [19].

With the ABAS-3, it is possible to obtain General Adaptive Composite (GAC) scores of the three adaptive domains and detailed assessments of individual skills, enabling more precise planning of therapeutic interventions and predicting the effects of difficulties that may affect the development or deterioration of functional skills. This, in turn, is of great importance in clinical practice [31,32].

The ABAS-3 can also act as a screening tool for identifying disabilities, particularly on the autism spectrum, where scores often indicate lower adaptive abilities. Research shows that social abilities are significantly associated with autism, regardless of intelligence level [7].

The authors of the tool [33] emphasize that the ABAS-3 generates scaled, norm-referenced and test-age equivalent scores, and the structure of skills and domains is consistent with AAIDD (American Association on Intellectual and Developmental Disabilities) and DSM-V (Diagnostic and Statistical Manual of Mental Disorders) guidelines, as well as strategies for working with a student with special educational needs, in line with the IDEA (Individuals With Disabilities Education Act) and RTI (Response to Intervention). In this way, the ABAS-3 supports individual and systemic intervention planning. Each subsequent measurement of adaptive skills is relevant to the impact of disorders or other conditions on the subject’s daily functioning. The ABAS-3 makes it possible to compare results obtained at different points in life to identify the most favorable learning conditions in children and to support independence at an older age. The information provided by the ABAS-3 is valuable for clinical decision-making and the design of individualized interventions.

Difference in perception between parents and teachers: The significant difference in ratings given by parents and teachers suggests that the two assessing groups have different perspectives on and experiences of interacting with children and adolescents on the autism spectrum. Parents may have a more optimistic or subjective view of their children’s adaptive skills, which would explain the higher average ratings in most categories. It is also possible that parents are more familiar with their child’s day-to-day functioning in their home environment, while teachers observe the child in the school context.

Areas of individual adaptive skills: The results show clear differences in the ratings given by parents and teachers in specific areas of adaptive skills. For example, parents rated skills related to self-determination, social skills and home life higher. This suggests that parents may be more aware of their child’s progress and challenges in these specific areas.

Regression model vs. adaptive skills: The regression analysis provided valuable information concerning the areas of adaptive skills that were key predictors of the General Adaptive Composite (GAC). For parents, self-care and leisure time were key, while teachers indicated a wider range of skills. This discrepancy may be due to differences in interactions with children and adolescents in different contexts—parents see their children in a more relaxed, home environment, while teachers observe them in a more structured school environment.

Adaptive domains: An important observation is that with regard to the conceptual adaptive domain, both groups—parents and teachers—agreed on key areas of individual adaptive behaviors. This may suggest that these concrete skills are more universal and evident both at home and at school. In terms of the social adaptive domain, the results suggest that skills related to engaging in and planning leisure time and establishing social interactions were most important to both parents and teachers. This indicates how important these skills are to the overall functioning of adolescents on the autism spectrum in a variety of settings.

(a) Adaptive domain in the conceptual area: Both parents and teachers identified socializing as a significant predictor. For parents, leisure time was also identified as a predictor. This may indicate the need for more attention to social interaction and leisure time to support the development of conceptual skills for individuals on the autism spectrum.

Social interactions, such as talking, playing together or participating in group therapy activities, can provide a key environment for learning and practicing conceptual skills. Such interactions can help people on the autism spectrum understand abstract ideas, interpret social cues and develop the ability to see the world from different perspectives.

In addition, rest and relaxation are essential to each person’s mental and physical health. Individuals on the autism spectrum may be particularly vulnerable to sensory overload or information-processing stress. Regular breaks, moments of calm and rest can help regenerate the mind, which translates into better cognitive functions and the ability to learn new conceptual skills.

Therefore, it is crucial that educators, therapists and caregivers pay special attention to promoting quality social interactions and providing appropriate leisure time for people on the autism spectrum. Supporting these aspects of their lives can significantly contribute to their overall development.

(b) Adaptive domain in the social area: Here, too, differences were found between predictors in the assessments provided by parents and teachers. For parents, social skills are key, while teachers identify communication and practical learning as important. This suggests that different contexts may influence perceptions of social skills, especially differences in perceptions and expectations in different environments.

The results suggest that the context in which a person on the autism spectrum operates has a significant impact on what skills are perceived as crucial. Social skills, such as participating in social events, interacting with peers or daily activities in a social setting, are an important part of each person’s life. For parents who see their children in everyday situations outside of school, being active and involved in the community can be a key indicator of their child’s adaptability.

Meanwhile, teachers who observe students in a more structured school environment may see skills related to communication and practical learning as crucial to success in this context. Communication, both verbal and non-verbal, is the foundation of effective social interactions in educational settings. Practical learning skills, such as organization, self-control and the ability to work in a group, are essential for successful functioning in the classroom.

This is why it is so important to take a holistic approach to assessing and supporting the social skills of individuals on the autism spectrum. A single perspective—whether that of the parent or the teacher—may not be sufficient to fully understand a person’s needs and abilities. Instead, integrating different perspectives and experiences can lead to a more comprehensive picture of a person’s adaptive skills and inform the most effective support strategies.

(c) Adaptive domain in the practical area: In the case of teachers, an additional predictor is leisure time, which may reflect differences in the assessment of practical skills in different environments and suggest that teaching and support methods may differ depending on the context. It may also point to the need for research on the impact of different environments on practical skills.

The fact that teachers identify leisure time as an additional predictor in the practical area underscores the variety of situations that individuals on the autism spectrum face in different environments. Leisure in the context of teachers can be understood not only as physical rest but also as the ability to manage stress, emotional regulation and the ability to organize leisure time independently. In a school environment, where the daily rhythm is predetermined and offers a variety of stimuli, people on the autism spectrum can face difficulties in adapting and finding time for recuperation. The fact that teachers recognize this need for rest shows their awareness of the challenges their students face. The diversity of contexts—home, school, community—affects the diversity of needs and adaptation strategies. What is crucial in one environment may not be so important in another. Therefore, teaching and support methods should be flexible and tailored to the specific situation. For instance, strategies to help manage stress at school may differ from those employed at home.

This finding also points to the importance of conducting further research on the impact of different environments on the development of practical skills for individuals on the autism spectrum. This research could provide valuable insights into how different environmental factors affect adaptive skills and what support strategies are most effective in different contexts. Understanding these dynamics is key to creating more effective support and education programs for individuals on the autism spectrum.

In summary, regression analysis revealed differences in key predictors of the General Adaptive Composite depending on the respondent group.

(a)Conceptual domain: Both parents and teachers identified socializing as a major predictor. Social interactions and moments of leisure are key to the development of conceptual skills.(b)Social domain: Parents emphasized social skills, while teachers focused on communication and practical learning. This indicates the diversity of expectations in different environments.(c)Practical domain: For teachers, the key predictor was leisure time, suggesting the need to adjust support strategies depending on the environment.

The obtained results highlight the importance of integrating the perspectives of both parents and teachers in assessing and supporting adolescents on the autism spectrum. Individual adaptive skills play a key role in different areas of adaptation, while their perception is determined by the respondent’s environmental context.

### 5.2. Summary

The analysis shows that while there are some differences in the assessments made by parents and teachers regarding the adaptive skills of adolescents on the autism spectrum, the two groups have valuable and complementary perspectives. Understanding these differences and similarities should support therapeutic work.

The ABAS-3 can be helpful to teachers, therapists and clinicians in areas such as:

-Assessment of adaptive skills.-Diagnosis and classification of disabilities and disorders.-Identification of strengths and weaknesses.-Documenting and monitoring progress.-Developing therapeutic plans.-Determining entitlement to disability benefits.-Assessing ability to lead an independent life or work.-Developing strategies to support functioning at home, school, work and in the community.-Planning interventions including teacher and family involvement.

However, such comprehensive functionality of the tool is only possible if it covers people from 0 to 89 years of age and the disability assessment systems and educational strategies are compatible with each other, as in the American system, which is often lacking in the Polish context so far.

The ABAS-3 tool is not equipped for individualized assessment of people on the autism spectrum. Detailed regression analysis for three adaptive domains in children and adolescents with autism spectrum disorders reveals a complex structure of predictors that varies in parents’ and teachers’ assessments and provides important insights. The conclusions drawn from this analysis are crucial for the following reasons.

They emphasize the diversity of impact of adaptive skills: There is no uniform pattern of impact, indicating the need for an individualized approach to intervention.

They point out the potential limitations of assessment tools: The need to account for differences in perception between parents and teachers and the potential shortcomings of tools such as the ABAS-3 underscore the need for further investigation and possible adaptation of tools to meet the specific needs of individuals on the autism spectrum.

They have significant implications for practice: These findings can inform individualized therapeutic and educational interventions, emphasizing the importance of a comprehensive and individualized approach to supporting children and adolescents on the autism spectrum.

In Table 6, we present the summary of important observations and their implications, both for practice and further research.

### 5.3. Relevance to Clinical and Educational Practice

The findings from this study have important implications for clinical, educational and intervention practice. Incorporating different perspectives in assessing the adaptive behaviors of individuals with ASD can lead to more balanced and targeted interventions, tailored to the individual needs and abilities of those diagnosed.

To summarize the practical recommendations for educational and clinical practice, several areas should be mentioned, such as, in particular, improving communication between teachers and parents, tailoring interventions to individual needs and creating an inclusive educational environment.

Regular meetings between parents and teachers; the creation of individual educational and therapeutic plans, which are jointly discussed and monitored by both teachers and parents; and communication training for both groups should be standard practice and not, as is currently the case in Poland, rather occasional in school reality.

An individualized approach to intervention is crucial in the education and therapy of people on the autism spectrum. ABAS-3 results make it possible to develop work plans based on the child’s strengths and weaknesses, which are identified by both teachers and parents in terms of adaptive skills in three domains (conceptual, social and practical).

It is also important to have an environment where students on the autism spectrum receive support programs that enable them to function better among their peers. For this, courses and workshops for teachers on the specifics of ASD and methods of working in a diverse group are essential, as well as engaging neurotypical students for the acceptance of neurodiversity.

Implementing the above recommendations requires systemic solutions, including increased funding for early intervention services that enable the development of adaptive skills from an early age. Research shows that adaptive skills tend to deteriorate with age in the absence of support. Adaptations of diagnostic tools such as ABAS-3 to the Polish educational system are also needed. Funding should be directed not only to education, but also to training for parents to help them better support the development of their children’s adaptive skills in the home environment.

## 6. Conclusions

### 6.1. A Brief Summary of the Study Results

The results presented above are valuable for understanding the complex patterns of adaptive skills in children and adolescents with ASD and can serve as a basis for more effective planning and implementation of educational and therapeutic interventions.

### 6.2. Limitations of the Study

Limitations of this present study refer to the following aspects.

As the ABAS-3 is not a tool specifically designed to assess people on the autism spectrum, this may result in a lack of full precision in identifying their specific adaptive needs and behaviors. The results may not fully reflect the real adaptive abilities of people with ASD. This may limit the applicability of the results to therapeutic work planning.

In this study, there were significant differences between parents‘ and teachers’ assessments of adaptability. Parents were more likely to rate their children higher, which may be a result of subjectivity and a natural tendency to see their child in a more positive light. Teachers, on the other hand, observe children in a more structured school environment, which may also influence their perception of adaptive behavior. Such discrepancies can have a significant impact in terms of planning therapeutic interventions. This, however, makes the relevance of regular communication between these groups all the more important in the process of assessing adaptive behavior and the supportive interventions carried out.

This study was conducted in Poland, which means that the possibility of generalizing the results and relating them to other cultural and social contexts is limited. Differences in predictors of adaptability were observed in the regression models, indicating the need for an individualized approach to assessment. Difficulties associated with uniformly assigning values to different predictors may limit the applicability of the results to all individuals with ASD, and over-generalization may lead to ineffective interventions. The use of the ABAS-3 without full adaptation to the specific characteristics of people with ASD points to the need to develop more precise assessment standards for this group. The lack of such standards may lead to interpretive ambiguities and make it difficult to use the results as a basis for standardized support programs.

The examination of relationships between demographic variables such as age is severely limited by the very design of the ABAS tool, in which each diagnostic score is recalculated in relation to age. Also, we did not choose to carry out analyses by gender because the group of girls was too small.

It would have been best to carry out additional analyses by establishing structural equation modeling techniques from the group of so-called causal interpretation methods based on statistical analysis of the data, called path analysis. This was not performed because the group sizes were too small. Path analysis measures not only direct interactions between variables but also indirect interactions through other variables, following a defined path. It makes it possible to move from more complex to less complex models by eliminating links between variables. In turn, the linkage structures are represented by a path diagram, similar to an activity network, which presents variables interconnected by lines indicating causal flows. At the same time, it is assumed that the construction of a personal diagram depicting the relationships between variables cannot be derived from prior empirical analysis but only from substantive knowledge of the population, where the causal relationships are asymmetric; in other words, there are no reciprocal causal relationships in the analyzed system of variables. Furthermore, the existence of “return” relationships between two variables via other variables is excluded.

### 6.3. Suggestions for Further Research

Potential areas for further research include the following.

Research on the specificity of the autism spectrum: To better understand the needs of people on the autism spectrum, it is suggested that research be conducted using tools and methods focused exclusively on this population. This may include qualitative research that will provide a deeper analysis of individual experiences and needs.

Comparing the perceptions of different groups: Given the differences in perception between parents and teachers, it is suggested that studies be conducted that compare and analyze these differences. This can help understand how different environments and relationships affect perceptions of adaptive skills.

Research on individualized interventions: Research on the design and evaluation of individualized therapeutic and educational interventions can provide valuable data on the best methods of supporting diversity in adaptive skills in this population.

Research on the impact of different environments: Given the potential impact of different environments on practical skills, it is worthwhile to conduct research that analyzes how different contexts (school, home, community) affect the development and assessment of these skills.

Research on the generalizability of results: To assess whether results can be generalized beyond the Polish context, international studies that compare results and methods between different cultures and educational systems can be considered.

Analysis of limitations of existing tools: Conducting a detailed analysis of the limitations of tools such as the ABAS-3, in the context of the specific needs of this population, can lead to further adaptation and improvement of assessment tools.

In summary, the suggested directions for further research emphasize the need for individualized, culturally appropriate and holistic approaches to the study and support of individuals on the autism spectrum. These suggestions aim to promote research that is more tailored to the specific needs and context of this specific group, taking into account both individual characteristics and the impact of different environments and relationships.

## Figures and Tables

**Figure 1 jcm-13-07607-f001:**
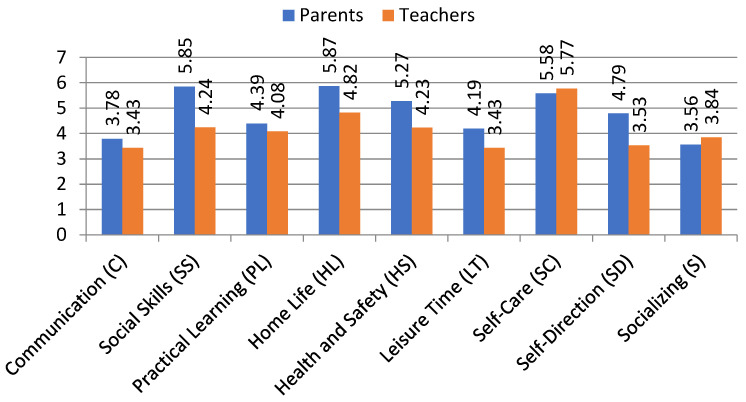
Key adaptive domains and skills in view of parents and teachers.

**Figure 2 jcm-13-07607-f002:**
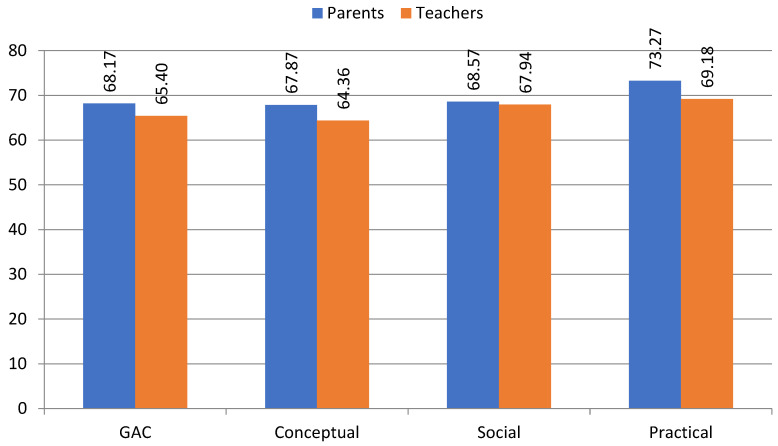
Ratings of key predictors of General Adaptive Composite (GAC) and adaptive domains.

**Table 1 jcm-13-07607-t001:** Sociodemographic data of the subjects in the study group.

Variable	Class	Total	
		n	%
Age groups	<10	45	45.45
	>9<16	40	40.40
	>15	14	14.15
Siblings	None	31	31.31
	One and more	68	68.69

**Table 2 jcm-13-07607-t002:** Comparison of parent and teacher ratings on key adaptive domains and skills.

Domain/Skill Area	Parent Avg. Score	Teacher Avg. Score	*p*-Value (Significance)	Effect Size
General Adaptive Composite (GAC)	68.17	65.40	0.031 (significant)	Weak
Conceptual Domain	67.87	64.36	0.001 (significant)	Weak
Practical Domain	73.27	69.18	0.002 (significant)	Weak
Social Skills	5.85	4.24	<0.001 (significant)	Moderate
Home/School Life	5.87/4.82	-	<0.001 (significant)	Moderate
Health and Safety	5.27	4.23	0.001 (significant)	Weak
Leisure Time	4.19	3.43	0.004 (significant)	Weak
Self-Direction	4.79	3.53	<0.001 (significant)	Moderate

**Table 3 jcm-13-07607-t003:** Key predictors of General Adaptive Composite (GAC) and adaptive domains.

Adaptive Domain	Predictors (Parent Ratings)	Predictors (Teacher Ratings)
General Adaptive Composite (GAC)	Self-Care (SC), Leisure Time (LT)	Self-Direction (SD), Social Skills (SS), SC, LT, Health and Safety (HS), Communication (C)
Conceptual Domain	Practical Learning (PL), Communication (C), Self-Direction (SD), Socializing (S)	Socializing (S), SD, C, PL
Social Domain	Social Skills (SS, inverse), LT, Socializing (S)	Socializing (S), LT, Communication (C, inverse)
Practical Domain	Home Life (HL), School Life (SL), SC, HS	HS, S, SS, SL, LT

**Table 4 jcm-13-07607-t004:** Comparison of ABAS-3 scores of studied children and adolescents with ASD as assessed by the parent (P) and the teacher (T).

Variable	Ave	SD	N	t °	df	*p*	d-Cohen
*General Adaptive Composite (GAC)*							
P GAC	68.17	16.96					
T GAC	65.40	14.19	86	2.199	85	0.031 *	W
*Adaptive Domains*							
P Conceptual	67.87	16.18					
T Conceptual	64.36	14.41	91	3.366	90	0.001 **	W
P Social	68.57	12.75					
T Social	67.94	13.13	88	0.486	87	0.628	–
P Practical	73.27	17.34					
T Practical	69.18	15.00	88	3.186	87	0.002 **	W
*Individual Adaptive Skill Areas*							
P Communication (C)	3.78	2.98					
T Communication	3.43	2.98	92	1.570	91	0.120	–
P Social Skills (SS)	5.85	3.90					
T Social Skills	4.24	3.48	92	4.853	91	0.000 ***	M
P Practical Learning (PL)	4.39	3.73					
T Practical Learning	4.08	3.45	92	1.151	91	0.253	–
P Home Life (HL)	5.87	3.17					
T School Life (SL)	4.82	2.83	92	3.665	91	0.000 ***	M
P Health and Safety (HS)	5.27	3.43					
T Health and Safety	4.23	2.84	90	3.322	89	0.001 **	W
P Leisure Time (LT)	4.19	3.03					
T Leisure Time	3.43	2.67	90	2.969	89	0.004 **	W
P Self-Care (SC)	5.58	3.27					
T Self-Care	5.77	3.34	91	−0.596	90	0.552	–
P Self-Direction (SD)	4.79	3.02					
T Self-Direction	3.53	2.40	92	4.994	91	0.000 ***	M
P Socializing (S)	3.56	2.83					
T Socializing	3.84	2.58	90	−1.197	89	0.234	–

*** *p* < 0.000; ** *p* < 0.01; * *p* < 0.05. W—weak effect; M—moderate effect.

**Table 5 jcm-13-07607-t005:** Individual adaptive skills areas as predictors of the General Adaptive Composite (GAC) and the three adaptive domains of the studied children and adolescents with ASD as assessed by the parent and the teacher.

	Predictors	b *	Stat. Error with b *	b	Stat. Error with b	t (162)	*p*
GAC—Parent	Intercept term			41.78	1.36	3.665	0.000 ***
	SC	0.37	0.06	1.88	0.32	5.816	0.000 **
	LT	0.25	0.07	1.36	0.38	3.579	0.001 **
R = 0.93; R2 = 0.85; *p* < 0.000 ***.							
GAC—Teacher	Intercept term			43.76	0.72	6.823	0.000 ***
	SD	0.17	0.06	0.95	0.35	2.730	0.008 **
	SS	0.13	0.06	0.52	0.25	2.040	0.044 *
	SC	0.13	0.04	0.53	0.18	2.980	0.004 **
	LT	0.17	0.05	0.91	0.28	3.249	0.002 **
	HS	0.14	0.05	0.70	0.23	3.037	0.003 **
	C	0.12	0.05	0.55	0.26	2.136	0.035 *
R = 0.97; R2 = 0.94; *p* < 0.000 ***.							
Conceptual—Parent	Intercept term			45.42	0.48	93.885	0.000 ***
	PL	0.41	0.03	1.78	0.14	13.064	0.000 ***
	C	0.38	0.03	2.05	0.14	14.497	0.000 ***
	SD	0.26	0.03	1.38	0.16	8.398	0.000 ***
	S	0.07	0.03	0.38	0.19	2.031	0.045 *
R = 0.99; R2 = 0.98; *p* < 0.000 ***.							
Conceptual—Teacher	Intercept term			44.57	0.27	163.621	0.000 ***
	S	0.42	0.02	1.80	0.08	23.261	0.000 ***
	SD	0.23	0.02	1.39	0.13	11.006	0.000 ***
	C	0.35	0.02	1.73	0.09	18.884	0.000 ***
	PL	0.03	0.01	0.19	0.08	2.247	0.027 *
R = 0.99; R2 = 0.99; *p* < 0.000 ***.							
Social—Parent	Intercept term			52.70	0.64	81.905	0.000 ***
	SS	−0.22	0.06	−0.70	0.18	−3.867	0.000 ***
	LT	0.60	0.06	2.49	0.23	1.887	0.000 ***
	S	0.61	0.05	2.71	0.22	12.255	0.000 ***
R = 0.95; R2 = 0.91; *p* < 0.000 ***.							
Social—Teacher	Intercept term			5.58	1.30	38.876	0.000 ***
	S	0.49	0.09	2.46	0.43	5.663	0.000 ***
	LT	0.44	0.10	2.17	0.49	4.425	0.000 ***
	C	−0.25	0.11	−1.11	0.47	−2.356	0.021 *
R = 0.88; R2 = 0.76; *p* < 0.000 ***.							
Practical—Parent	Intercept term			42.27	0.55	77.200	0.000 ***
	HL	0.26	0.03	1.43	0.17	8.509	0.000 ***
	SL	0.30	0.03	1.34	0.11	11.985	0.000 ***
	SC	0.30	0.03	1.62	0.14	11.191	0.000 ***
	HS	0.21	0.03	1.04	0.13	7.818	0.000 ***
R = 0.99; R2 = 0.97; *p* < 0.000 ***.							
Practical—Teacher	Intercept term			44.32	0.38	117.291	0.000 ***
	HS	0.30	0.02	1.55	0.11	13.843	0.000 ***
	S	0.31	0.02	1.41	0.09	15.137	0.000 ***
	SS	0.28	0.03	1.21	0.13	9.157	0.000 ***
	SL	0.18	0.02	0.95	0.13	7.437	0.000 ***
	LT	0.06	0.02	0.36	0.13	2.774	0.007 **
R = 0.99; R2 = 0.98; *p* < 0.000 ***.							

Legend: C = Communication, LT = Leisure Time, HS = Health and Safety, PL = Practical Learning, S = Socializing, SC = Self-Care, SD = Self-Determination, SL = School Life, SS = Social Skills. *** *p* < 0.000; ** *p* < 0.01; * *p* < 0.05.

**Table 6 jcm-13-07607-t006:** Summary of significant observations and implications.

Observation Type	Key Insights
Differences in Assessments	Parents rated children higher than teachers across most domains, suggesting potential subjectivity.
Significant Divergences	Notable differences were seen in ratings of self-determination, social skills and home life.
Converging Scores	Both parents and teachers identified similar predictors for the conceptual domain (communication, PL, SD, S).
Predictor Patterns	Parents’ GAC predictors were limited to two areas (SC, LT), while teachers had a broader range.
Social Domain Perceptions	Differences were observed in communication and social skills as predictors in the social domain.
Educational Implications	The need for individualized support strategies and better parent–teacher communication was highlighted.

## Data Availability

Additional raw data available on demand from authors.

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
