# Peer review of "Predictors of Adaptive Behaviors in Individuals on the Autism Spectrum as Assessed by Teachers and Parents: An Analysis Based on ABAS-3"

_jcm, 2024, doi:10.3390/jcm13247607_

Round 1
Reviewer 1 Report
Comments and Suggestions for Authors
This topic is important because it can help identify factors that contribute to positive outcomes for individuals with autism spectrum disorder (ASD). By understanding these predictors, researchers and clinicians can develop targeted interventions to enhance adaptive behaviors, ultimately improving the quality of life for individuals with ASD. This research can also inform educational and social policies to support the needs of individuals with ASD better, promoting their inclusion and independence.
To improve this manuscript I suggest the following points for consideration:
Introduction
· Clearly state the main research question or hypothesis. Highlight the unique contribution of your study to the existing literature, especially regarding the comparative analysis of parent and teacher ratings using the ABAS-3.
· I suggest focusing on relevant studies: Prioritize studies that directly address the relationship between parent and teacher ratings of adaptive behaviors in individuals with ASD, especially those using the ABAS-3. I also suggest critically analyzing the literature, identifying gaps in the literature, and explaining how your study will fill those gaps. I am sure that this is done by highlighting the limitations of previous research. Discuss potential biases or methodological limitations in previous studies that the present study aims to address.
· Organize the literature review by theme (e.g., predictors of adaptive behaviors, the role of parents and teachers, the use of ABAS-3) rather than chronologically.
Methods
To improve this section I suggest:
· Specify the criteria used to select participants, such as diagnostic criteria, age range, and severity of ASD.
· Explain how participants were recruited and how informed consent was obtained.
· Provide More Details on the ABAS-3 Administration. Describe the specific procedures for administering the ABAS-3 to parents and teachers, including training, supervision, and data collection methods.
· Discuss the reliability and validity of the ABAS-3 in the Polish context.
· I think that the authors need to justify the use of statistical analyses. I suggest explaining why authors chose the specific statistical tests (e.g., t-test and multivariate stepwise analysis) to analyze the data. I also suggest describing the assumptions underlying these tests and how the authors ensured they were met.
Results
The results section is well-written and effectively communicates the key findings of the study. However, I have the following suggestions to improve this section:
Consider Additional Analyses for the data. Explore the relationship between specific demographic variables (e.g., age, gender, severity of ASD) and adaptive behavior scores. Conduct further analyses to identify factors that may moderate or mediate the relationship between parent and teacher ratings.
Discussion
This section in the manuscript was mistakenly mentioned as the results! I suggest changing it to “Discussion”. My other recommendations are:
Strengthen the Theoretical Framework and connect to Existing Literature. To be more precise and more explicitly connect the findings to existing theories and research on autism spectrum disorder and adaptive behavior.
Discuss the Implications for theoretical understanding. I suggest considering how the findings contribute to a deeper understanding of the factors influencing adaptive behavior in individuals with ASD.
Delve deeper into the practical implications and provide more concrete recommendations for educational and clinical practice, such as strategies for improving communication between parents and teachers, tailoring interventions to individual needs, and creating inclusive environments.
Discuss the potential policy implications of the findings, such as the need for increased funding for early intervention services and inclusive education.
I also suggest addressing the limitations more explicitly. Discuss the impact of limitations and explain how the study's limitations may have influenced the findings and the generalizability of the results.
Author Response
Thank you very much for all your suggestions, which we believe have helped to remedy the shortcomings of the article. We have taken all of them into account, except the suggestion for additional analyses. We believe they are relevant, but in our response we explain the reasons why we could not perform them and how this could be done in future studies.
Below is our point-by-point response. In the attached word file, black font shows the Reviewer's comments, orange font our responses.
Reviewer 1 comments
Introduction
- Clearly state the main research question or hypothesis. Highlight the unique contribution of your study to the existing literature, especially regarding the comparative analysis of parent and teacher ratings using the ABAS-3.
We have included the following excerpt in the introduction:
Our study addresses the important question, whether there are differences in the areas of individual adaptive behaviors as predictors of the General Adaptive Composite (GAC) and the three adaptive domains (conceptual, social, and practical) of children and adolescents on the autism spectrum, as assessed by parents and teachers? It provides a comparative analysis of parent and teacher ratings of adaptive behaviors in individuals with Autism Spectrum Disorder (ASD) using the Adaptive Behavior Assessment System, Third Edition (ABAS-3), therefore contributing to the existing literature by demonstrating how the ABAS-3 can be used to analyze adaptive behaviors in different contexts and by illustrating the variability in assessments between key stakeholders. The study highlights discrepancies between parent and teacher ratings across various adaptive domains and skills, which is a currently unfilled gap in this area, emphasizing the need to incorporate multiple perspectives for a holistic understanding of adaptive behaviors in ASD. Due to the fact it was conducted in Poland, the research underscores cultural and methodological challenges in using the ABAS-3 for assessing individuals with ASD, revealing the limitations of existing tools and the need for localized adaptations.
- I suggest focusing on relevant studies: Prioritize studies that directly address the relationship between parent and teacher ratings of adaptive behaviors in individuals with ASD, especially those using the ABAS-3. I also suggest critically analyzing the literature, identifying gaps in the literature, and explaining how your study will fill those gaps. I am sure that this is done by highlighting the limitations of previous research. Discuss potential biases or methodological limitations in previous studies that the present study aims to address.
The reviewer's comments are in line with our approach to the study, and we have therefore included detailed references to the literature that demonstrate existing research gaps and point out the limitations of previous work. Our study fills this gap by offering analyses based on a larger sample and addressing potential sources of bias and methodological limitations. Previous studies have often been characterised by a small research sample, limited scope of analysis or failure to take into account the concordance rate between informants. Our study takes a step towards overcoming these limitations by providing new data on the relationship between parent and teacher ratings using the ABAS-3 tool. We have included the following excerpt in the introduction:
‘To date, studies of the adaptive behaviour of individuals with ASD have rarely analysed the relationship between parent and teacher ratings (Kenworthy, L., Case, L., Harms, M.B. et al., 2010). Where such analysis has been undertaken, research has typically been conducted on relatively small samples (Arciuli J, Stevens K, Trembath D, Simpson IC, 2013). Assessments by different groups of informants are considered the gold standard in the diagnosis and assessment of children's functioning. Despite this, few studies have focused on examining the concordance of assessments of adaptive functioning in adolescents with ASD, particularly using the ABAS-3 tool. Our study, conducted on a large and diverse group of adolescents with ASD, examined the concordance of parent and teacher assessments of both adaptive functioning and ASD-specific symptoms at different time points (Dickson, K.S., Suhrheinrich, J., Rieth, S.R. et al., 2018).’
- Organize the literature review by theme (e.g., predictors of adaptive behaviors, the role of parents and teachers, the use of ABAS-3) rather than chronologically.
We re-arranged the literature review section as suggested:
Predictors of Adaptive Behaviors
Research on predictors of adaptive behaviors in individuals with autism spectrum disorder (ASD) reveals different findings, reflecting the complexity of this domain. Some studies indicate that variables such as age, gender, family income, or years of formal education are not significant predictors of adaptive behaviors (Klin et al., 2007; Kenworthy et al., 2010). By contrast, other studies highlight age and gender as significant factors influencing deficits in daily living skills (Duncan & Bishop, 2015).
Age emerges as a particularly strong predictor of communication and social skills (Pugliese et al., 2015), while motor skills are consistently identified as key predictors of adaptive behaviors (Bo, 2015; Fears et al., 2022). Differences in motor coordination, particularly hand coordination, may underlie variations in adaptive functioning (Fears et al., 2022). Furthermore, intelligence quotient (IQ) has been found to be a poor predictor of adaptive functioning in children with ASD, who often score lower in adaptive behaviors than their IQ would suggest (Kanne et al., 2011). Notably, adaptive skills in children with ASD tend to decline with age, suggesting a developmental trajectory distinct from typically developing peers (Klin et al., 2007).
Additionally, early diagnosis of executive function deficits has been identified as a potential predictor of later adaptive behaviors, underscoring the importance of early intervention (Pugliese et al., 2015). Mental age in infancy also correlates with trajectories in communication, daily living skills, and socialization (Pathak et al., 2019).
The Role of Parents and Teachers
Studies exploring adaptive functioning often highlight the discrepancies between parent and teacher ratings, revealing context-dependent variations. For example, a study using ABAS-3 found that teachers consistently provided higher adaptive behavior ratings compared to parents, suggesting better functioning in structured school environments (Jordan et al., 2019). These discrepancies may reflect the differing perspectives and expectations of parents and teachers or variations in behavior across home and school settings. Studies indicate that while parent and teacher ratings are moderately correlated, teachers often rate adaptive skills higher than parents. For instance, Stevens et al. (2022) found that teachers rated youth with ASD and intellectual disabilities as having higher adaptive skills compared to parent ratings, with autism symptomatology significantly predicting these discrepancies
While there is limited research directly comparing parent and teacher ratings, existing studies suggest that differences in scores could indicate unique adaptive behavior profiles in various contexts. Understanding these variations is crucial for tailoring interventions that address both home and school environments (Jordan et al., 2019).
Use of ABAS-3
The ABAS-3 has been increasingly used in assessing adaptive behaviors in individuals with ASD, although it was not specifically designed for this population. Compared to the Vineland Adaptive Behavior Scales (VABS), ABAS-3 offers a broader focus on daily living and practical skills but may lack specificity in assessing the unique challenges faced by individuals with ASD. For example, while VABS is frequently used to measure adaptive behavior trajectories, particularly in large-scale studies (e.g., Pathak et al., 2019), ABAS-3 provides a more standardized framework for examining differences across informants, such as parents and teachers.
However, the limitations of using ABAS-3 in ASD populations should be addressed. The tool’s general design may introduce biases, as it does not account for the distinct developmental profiles of individuals with ASD. For instance, the developmental decline in adaptive skills with age, as observed in ASD populations (Klin et al., 2007; Pugliese et al., 2015), may not be fully captured by ABAS-3's scoring system. Future studies should consider these limitations and explore how ABAS-3 findings align with those derived from VABS to ensure comprehensive assessments.
In the Polish context, it is worth noting that until the 1990s, professionals in Poland had very limited access to materials and tools for diagnosing disabilities, including those related to adaptive behaviors. The only tools used in Poland until recently for assessing adaptive behaviors were B. Doll’s Social Maturity Scale and Witkowski’s PAC (Progress Assessment Chart). For this reason, work on adaptations and the promotion of the use of new tools in this context is so important.
Methods
To improve this section I suggest:
- Specify the criteria used to select participants, such as diagnostic criteria, age range, and severity of ASD.
We have combined the response to this and the following point into one paragraph below.
- Explain how participants were recruited and how informed consent was obtained.
We added a following paragraphs:
As part of the initial work on the adaptation of ABAS-3 in Poland, a group of individuals on the autism spectrum was included in the study. All subjects had a diagnosis of autism according to the ICD-10 criteria, which is still in force in Poland. According to these criteria, the severity of symptoms is not specified. The age range is clearly defined for the ABAS tool, so chronological age was decisive in the selection of study participants. The study participants came from 28 different centres and educational facilities, including integrated and special kindergartens, special education centres and integration schools.
The research was carried out as part of the functional diagnosis of the study subjects at the above-mentioned centres. Both the carers and the subjects themselves were informed that they could withdraw their consent to take part in the study at any time without stat-ing a reason and without any consequences, and that all personal data would be anonymised, meaning that the carer's data and the child's data could not be linked to the results of the study.
- Provide More Details on the ABAS-3 Administration. Describe the specific procedures for administering the ABAS-3 to parents and teachers, including training, supervision, and data collection methods.
We added a following paragraph:
The interviewers were appropriately trained and had adequate knowledge and expe-rience in basic educational and psychological assessment and test interpretation. There was a person in charge of coordinating the completion of the forms at each facility. Par-ents, teachers, family members and carers of the subjects and the subjects themselves completed the form using printed questionnaires.
- Discuss the reliability and validity of the ABAS-3 in the Polish context.
We added a following paragraph:
The analysis of the reliability of the ABAS-3 tool in the Polish context was performed by Wojciech Otrębski, Ewa Domagała-Zysk and Agnieszka Sudoł [22] on the basis of the results obtained in the study of the standardisation sample. Internal consistency, standard errors of measurement, stability and consistency between different worksheets concerning the same individuals were determined. The reliability coefficient ranges from 0.86 to 0.99 depending on the worksheet, the lowest being in the age range 0;2-0;11. The recalculated scores for adaptive functions correlate moderately strongly with each other.
- I think that the authors need to justify the use of statistical analyses. I suggest explaining why authors chose the specific statistical tests (e.g., t-test and multivariate stepwise analysis) to analyze the data. I also suggest describing the assumptions underlying these tests and how the authors ensured they were met.
We have introduced passages to justify the choice of specific statistical tests:
A test-t was performed where the distributions in the populations are assumed to be approximately normal. This is important for small samples (n<30), because for larger samples the test is robust to moderate deviations from this assumption. Furthermore, the test requires the assumption of equality of variance in both populations.
Associations between variables were established using stepwise multiple regression analysis, which allows the arrangement of explanatory variables that are significant in explaining the dependent variable to be identified and the strength of the association be-tween them to be estimated in the regression model so constructed. The essence of step-wise regression is to leave a minimum set of explanatory variables in the regression function model, while maximising the coefficient of determination and minimising the mean square of the deviations from the regression model, in order to identify the dimensions of those variables that most often combine with each other and those that are in conflict with each other, i.e. interacting with the phenomenon under study in a stimulating or inhibiting way. Stimulants are variables with positive regression parameter values, while des-timulants are variables with negative parameters. On the other hand, statistically insig-nificant variables, so-called neutral variables, do not participate in the analysis of the correlation under study. At the same time, multiple regression coefficients have different units of measurement, so they cannot be used to directly compare the strength of the association between individual explanatory variables and the explained variable.
Results
The results section is well-written and effectively communicates the key findings of the study. However, I have the following suggestions to improve this section:
- Consider Additional Analyses for the data. Explore the relationship between specific demographic variables (e.g., age, gender, severity of ASD) and adaptive behavior scores. Conduct further analyses to identify factors that may moderate or mediate the relationship between parent and teacher ratings.
Thank you for this comment. We also believe that further work and analysis in this area is needed. In the Limitations section, we described why it was not possible to perform these analyses for our study and how they could be performed in further studies.
The examination of relationships between demographic variables such as age is severely limited by the very design of the ABAS tool, in which each diagnostic score is re-calculated in relation to age. Also we did not choose to carry out analyses by gender be-cause the group of girls was too small.
It would have been best to carry out additional analyses by establishing structural equation modelling techniques, from the group of so-called causal interpretation methods based on statistical analysis of the data, called path analysis. This was not done because the group sizes were too small. Path analysis measures not only direct interactions be-tween variables, but also indirect interactions through other variables, following a defined path. It makes it possible to move from more complex to less complex models by eliminating links between variables. In turn, the linkage structures are represented by a path diagram, similar to an activity network, which presents variables interconnected by lines indicating causal flows. At the same time, it is assumed that the construction of a personal diagram depicting the relationships between variables cannot be derived from prior empirical analysis, but only from substantive knowledge of the population, where the causal relationships are asymmetric, in other words, there are no reciprocal causal relationships in the analysed system of variables. Furthermore, the existence of ‘’return‘’ relationships between two variables via other variables is excluded.
Discussion
- This section in the manuscript was mistakenly mentioned as the results! I suggest changing it to “Discussion”.
We have changed the name of the section to Discussion.
My other recommendations are:
- Strengthen the Theoretical Framework and connect to Existing Literature. To be more precise and more explicitly connect the findings to existing theories and research on autism spectrum disorder and adaptive behavior.
We have done this by adding a discussion in the paragraph below:
The ABAS-3 is distinguished by the simplicity of conducting surveys, which is due to the short time required to complete the questionnaire. This is particularly useful when working with large groups (cf. Mahendiran T. et al., 2019; Košmrlj, L., 2018).
The tool also allows for the analysis of differences in assessments stemming from varying data sources, such as a teacher and a parent, or an assessment made by the researcher itself. Such comparisons support clinical practice and therapeutic intervention by providing insights into the diverse perspectives of the examinee's environment (Jordan A.K. et al., 2019).
With the ABAS-3, it is possible to obtain an general adaptive component (GAC), scores of the three adaptive domains, and detailed assessments of individual skills, enabling more precise planning of therapeutic interventions and predicting the effects of difficulties that may affect the development or deterioration of functional skills. This, in turn, is of great importance in clinical practice (Dégeilh F., Bernier A., Gravel J., Beauchamp M.H., 2018; Ricci M.F., 2018).
The ABAS-3 can also act as a screening tool for identifying disabilities, particularly on the autism spectrum, where scores often indicate lower adaptive abilities. Research shows that social abilities are significantly associated with autism, regardless of intelligence level (Kenworthy L. et al., 2010).
The authors of the tool (Harrison P., Oakland T., 2015) emphasise that the ABAS-3 generates scaled, norm-referenced and test-age equivalent scores, and the structure of skills and domains is consistent with AAIDD (American Association on Intellectual and Developmental Disabilities), DSM-V (Diagnostic and Statistical Manual of Mental Disorders) guidelines, as well as strategies for working with a student with special educational needs, in line with IDEA (Individuals With Disabilities Education Act) and RTI (Response to Intervention). In this way, the ABAS-3 supports individual and systemic intervention planning. Each subsequent measurement of adaptive skills is relevant to the impact of disorders or other conditions on the subject's daily functioning. The ABAS-3 makes it possible to compare results obtained at different points in life to identify the most favourable learning conditions in children and to support independence in older age. The information provided by the ABAS-3 is valuable for clinical decision-making and the design of individualised intervention.
- Discuss the Implications for theoretical understanding. I suggest considering how the findings contribute to a deeper understanding of the factors influencing adaptive behavior in individuals with ASD.
We have done this by adding a discussion in the paragraph below:
The ABAS-3 can be helpful to teachers, therapists and clinicians in areas such as:
- assessment of adaptive skills,
- diagnosis and classification of disabilities and disorders,
- identification of strengths and weaknesses,
- documenting and monitoring progress,
- developing therapeutic plans,
- determining entitlement to disability benefits,
- assessing ability to lead an independent life or work,
- developing strategies to support functioning at home, school, work and in the community,
- planning interventions including teacher and family involvement.
However, such comprehensive functionality of the tool is only possible if it covers people from 0 to 89 years of age and the disability assessment systems and educational strategies are compatible with each other, as in the American system, which is often lack-ing in the Polish context so far.
- Delve deeper into the practical implications and provide more concrete recommendations for educational and clinical practice, such as strategies for improving communication between parents and teachers, tailoring interventions to individual needs, and creating inclusive environments. Discuss the potential policy implications of the findings, such as the need for increased funding for early intervention services and inclusive education.
We discussed the practical implications of our study in more detail:
To summarise the practical recommendations for educational and clinical practice, several areas should be mentioned. In particular, improving communication between teachers and parents, tailoring interventions to individual needs and creating an inclusive educational environment.
Regular meetings between parents and teachers, the creation of individual educational and therapeutic plans, which are jointly discussed and monitored by both teachers and parents, and communication training for both groups should be standard practice and not, as is currently the case in Poland, rather occasional in school reality.
An individualised approach to intervention is crucial in the education and therapy of people on the autism spectrum. ABAS-3 results make it possible to develop work plans based on the child's strengths and weaknesses, which are identified by both teachers and parents in terms of adaptive skills in three domains (conceptual, social and practical).
It is also important to have an environment where students on the autism spectrum receive support programmes that enable them to function better among their peers. For this, courses and workshops for teachers on the specifics of ASD and methods of working in a diverse group are essential, as well as engaging neurotypical students for the acceptance of neurodiversity.
Implementing the above recommendations requires systemic solutions, including increased funding for early intervention services that enable the development of adaptive skills from an early age. Research shows that adaptive skills tend to deteriorate with age in the absence of support. Adaptations of diagnostic tools such as ABAS-3 to the Polish educational system are also needed. Funding should be directed not only to education, but also to training for parents to help them better support the development of their children's adaptive skills in the home environment.
- I also suggest addressing the limitations more explicitly. Discuss the impact of limitations and explain how the study's limitations may have influenced the findings and the generalizability of the results.
The limitations were re-written according to this comment:
Limitations of the present study refer to the following aspects.
As the ABAS-3 is not a tool specifically designed to assess people on the autism spectrum, this may result in a lack of full precision in identifying their specific adaptive needs and behaviours. The results may not fully reflect the real adaptive abilities of people with ASD. This may limit the applicability of the results to therapeutic work planning.
In the study, there were significant differences between parents‘ and teachers’ assessments of adaptability. Parents were more likely to rate their children higher, which may be a result of subjectivity and a natural tendency to see their child in a more positive light. Teachers, on the other hand, observe children in a more structured school environ-ment, which may also influence their perception of adaptive behaviour. Such discrepancies can have a significant impact in terms of planning therapeutic interventions. Which, however, makes the relevance of regular communication between these groups all the more important in the process of assessing adaptive behaviour and the supportive interventions carried out.
The study was conducted in Poland, which means that the possibility of generalising the results and relating them to other cultural and social contexts is limited. Differences in predictors of adaptability were observed in the regression models, indicating the need for an individualised approach to assessment. Difficulties associated with uniformly assign-ing values to different predictors may limit the applicability of the results to all individuals with ASD, and over-generalization may lead to ineffective interventions. The use of the ABAS-3 without full adaptation to the specific characteristics of people with ASD points to the need to develop more precise assessment standards for this group. The lack of such standards may lead to interpretive ambiguities and make it difficult to use the results as a basis for standardised support programmes.
The examination of relationships between demographic variables such as age is se-verely limited by the very design of the ABAS tool, in which each diagnostic score is re-calculated in relation to age. Also we did not choose to carry out analyses by gender be-cause the group of girls was too small.
It would have been best to carry out additional analyses by establishing structural equation modelling techniques, from the group of so-called causal interpretation methods based on statistical analysis of the data, called path analysis. This was not done because the group sizes were too small. Path analysis measures not only direct interactions be-tween variables, but also indirect interactions through other variables, following a defined path. It makes it possible to move from more complex to less complex models by elimi-nating links between variables. In turn, the linkage structures are represented by a path diagram, similar to an activity network, which presents variables interconnected by lines indicating causal flows. At the same time, it is assumed that the construction of a personal diagram depicting the relationships between variables cannot be derived from prior empirical analysis, but only from substantive knowledge of the population, where the causal relationships are asymmetric, in other words, there are no reciprocal causal relationships in the analysed system of variables. Furthermore, the existence of ‘’return‘’ relationships between two variables via other variables is excluded.

Reviewer 2 Report
Comments and Suggestions for Authors
The manuscript presents important findings, but methodological and interpretative improvements are needed for publication. Below, detailed feedback is provided, section by section, with specific recommendations to address key issues:
Major
1. Methods: the authors have not specified the recruitment setting for the participants (e.g., outpatient clinics, schools, etc.) nor mentioned whether parental/guardian informed consent was obtained for the study. This information is critical for assessing the study’s ethical compliance. Therefore, authors should specify the recruitment setting for participants and, most importantly, confirm that informed consent was obtained from their parents or guardians.
Minor
1. Background: the literature review would benefit from further discussion on why ABAS-3 was chosen over tools, such as the Vineland Adaptive Behavior Scales. Additionally, the limitations of using ABAS-3, a tool not specifically developed for individuals with ASD, should be briefly addressed to provide readers with context on potential biases. Lastly, in the background section, lines 156-158, the authors state, “Scores from the ABAS-3 are used in the diagnosis of developmental disorders, identification of functional limitations, treatment planning, and research.” However, to support this statement, the authors currently cite only their own work. the current reviewer believes that it would strengthen the argument, and consequently the paper, to include additional references from independent sources.
2. Results: the presentation of results is dense and could benefit from clearer organization, such as through summary tables or graphs to facilitate interpretation; regarding the provided tables, this reviewer believes that Table 3 would benefit from a legend that defines the abbreviations used for ABAS-3 domains (e.g., HS for Health and Safety, HL for Home Life, SC for Self-Care), thus increasing clarity for readers who may not be familiar with these abbreviations. Moreover, the paper presents two consecutive “Results” sections (section 4 and 5), both presenting data and their interpretation. Additionally, an analysis by age and gender would enhance the depth of the findings, as adaptive behaviors may vary across these demographic factors. Therefore, the reviewer suggests to: 1) reformat and merge the results sections to improve clarity; authors should consider the use of summary tables or graphical representations; 2) Include analyses of differences by age and gender, if possible, to provide additional insights.
3. Discussion: following up on the comment for the Results section, the paper would benefit from a clearer distinction between results and their interpretation in the discussion. A more structured separation would enhance clarity, allowing readers to understand the raw findings before engaging with the authors' interpretations and implications.
Author Response
Answer to Reviewer 2
Thank you very much for all your suggestions, which we believe have helped to remedy the shortcomings of the article. We have taken all of them into account, except the suggestion for additional analyses. We believe they are relevant, but in our response we explain the reasons why we could not perform them and how this could be done in future studies.
Below is our point-by-point response. In the attached file, black font shows the Reviewer's comments, orange font our responses.
Reviewer 2 comments
Major
- Methods: the authors have not specified the recruitment setting for the participants (e.g., outpatient clinics, schools, etc.) nor mentioned whether parental/guardian informed consent was obtained for the study. This information is critical for assessing the study’s ethical compliance. Therefore, authors should specify the recruitment setting for participants and, most importantly, confirm that informed consent was obtained from their parents or guardians.
We have clarified these issues by adding paragraphs in the appropriate places in the section:
As part of the initial work on the adaptation of ABAS-3 in Poland, a group of indi-viduals on the autism spectrum was included in the study. All subjects had a diagnosis of autism according to the ICD-10 criteria, which is still in force in Poland. According to these criteria, the severity of symptoms is not specified. The age range is clearly defined for the ABAS tool, so chronological age was decisive in the selection of study participants. The study participants came from 28 different centres and educational facilities, including integrated and special kindergartens, special education centres and integration schools.
The research was carried out as part of the functional diagnosis of the study subjects at the above-mentioned centres. Both the carers and the subjects themselves were informed that they could withdraw their consent to take part in the study at any time without stat-ing a reason and without any consequences, and that all personal data would be anony-mised, meaning that the carer's data and the child's data could not be linked to the results of the study.
Minor
- Background: the literature review would benefit from further discussion on why ABAS-3 was chosen over tools, such as the Vineland Adaptive Behavior Scales. Additionally, the limitations of using ABAS-3, a tool not specifically developed for individuals with ASD, should be briefly addressed to provide readers with context on potential biases.
We have introduced the following paragraph explaining the use of this tool and int limitations: We have also rewritten and added some paragraphs, in line with Reviewer 1's comment, in of the literature review section.
‘Also the Vineland Adaptive Behaviour Scales tool is only partially translated and pre-adapted in Poland. Due to the advanced adaptation process of the ABAS-3, this tool was chosen. The main limitation of the ABAS-3 tool currently in Poland is its adaptation only to 20-year-old subjects and the lack of studies of subjects on the autism spectrum. Comprehensive measures are only possible when the disability assessment system or the system for developing strategies for working with a student on the autism spectrum are compatible with each other, at least to the extent that the American system provides for it. It is to be hoped that the ABAS-3 issued in Poland will be extended with a full adaptation as soon as possible, and that the Polish health, education and social welfare systems will use the same jointly developed tools for diagnosis and planned support. Only then will the ABAS-3 be fully adapted to the purpose for which the assessment is being prepared, and the results obtained will be useful and properly applied. The original tool offers such possibilities, but requires appropriate adaptation for people on the autism spectrum.’
- Lastly, in the background section, lines 156-158, the authors state, “Scores from the ABAS-3 are used in the diagnosis of developmental disorders, identification of functional limitations, treatment planning, and research.” However, to support this statement, the authors currently cite only their own work. the current reviewer believes that it would strengthen the argument, and consequently the paper, to include additional references from independent sources.
To better substantiate the statement that ABAS-3 scores are used in the diagnosis of developmental disorders, identification of functional limitations, therapy planning and research, we have additionally included the following papers in the text:
‘Research highlights the utility of the ABAS-3 tool in evaluating adaptive functioning in individuals with autism. Tamm, Day, and Duncan (2022) demonstrate that ABAS-3 effectively documents adaptive behavior profiles and identifies discrepancies between IQ and real-life skills, making it valuable for diagnosis and therapy planning, while Colantuono et al. (2023) underscore its effectiveness in capturing long-term changes in daily functioning across domains like self-care and health in neurodiverse populations, including autism.’
- Results: the presentation of results is dense and could benefit from clearer organization, such as through summary tables or graphs to facilitate interpretation;
As suggested by the reviewers, in order to present the data in a clearer way, we have added summary tables and graphs showing key results.
- regarding the provided tables, this reviewer believes that Table 3 would benefit from a legend that defines the abbreviations used for ABAS-3 domains (e.g., HS for Health and Safety, HL for Home Life, SC for Self-Care), thus increasing clarity for readers who may not be familiar with these abbreviations.
As suggested we added a legend to the table.
- Moreover, the paper presents two consecutive “Results” sections (section 4 and 5), both presenting data and their interpretation. Additionally, an analysis by age and gender would enhance the depth of the findings, as adaptive behaviors may vary across these demographic factors. Therefore, the reviewer suggests to: reformat and merge the results sections to improve clarity; authors should consider the use of summary tables or graphical representations;
We have tried to structure the presentation of the results to some degree, separated the Discussion section and, as suggested by the Reviewer, added summary tables and charts presenting key findings.
- Include analyses of differences by age and gender, if possible, to provide additional insights.
Thank you for this comment. We also believe that further work and analysis in this area is needed. In the Limitations section, we described why it was not possible to perform these analyses for our study and how they could be performed in further studies.
The examination of relationships between demographic variables such as age is severely limited by the very design of the ABAS tool, in which each diagnostic score is re-calculated in relation to age. Also we did not choose to carry out analyses by gender be-cause the group of girls was too small.
It would have been best to carry out additional analyses by establishing structural equation modelling techniques, from the group of so-called causal interpretation methods based on statistical analysis of the data, called path analysis. This was not done because the group sizes were too small. Path analysis measures not only direct interactions be-tween variables, but also indirect interactions through other variables, following a defined path. It makes it possible to move from more complex to less complex models by eliminating links between variables. In turn, the linkage structures are represented by a path diagram, similar to an activity network, which presents variables interconnected by lines indicating causal flows. At the same time, it is assumed that the construction of a personal diagram depicting the relationships between variables cannot be derived from prior empirical analysis, but only from substantive knowledge of the population, where the causal relationships are asymmetric, in other words, there are no reciprocal causal relationships in the analysed system of variables. Furthermore, the existence of ‘’return‘’ relationships between two variables via other variables is excluded.
- Discussion: following up on the comment for the Results section, the paper would benefit from a clearer distinction between results and their interpretation in the discussion. A more structured separation would enhance clarity, allowing readers to understand the raw findings before engaging with the authors' interpretations and implications.
We have addressed this comment by separating the Discussion section and by introducing summarizing tables and graphs showing the key raw results.

Round 2
Reviewer 1 Report
Comments and Suggestions for Authors
I appreciate the authors’ effort in addressing the raised issues by thoroughly considering the comments and suggestions. The updated version is an improvement and contributes to the existing literature in the field. I recommend publication in its current format.
Reviewer 2 Report
Comments and Suggestions for Authors
Authors have improved the manuscript as requested, this reviewer does not have any further request